# The Interaction of Innate and Adaptive Immunity and Stabilization of Mast Cell Activation in Management of Infusion Related Reactions in Patients with Fabry Disease

**DOI:** 10.3390/ijms21197213

**Published:** 2020-09-29

**Authors:** Renuka P. Limgala, Jaqueline Fikry, Vasudha Veligatla, Ozlem Goker-Alpan

**Affiliations:** Translational Research Unit, Lysosomal and Rare Disorders Research and Treatment Center, Fairfax, VA 22030, USA; jfikry@ldrtc.org (J.F.); veligatlavasudha@gmail.com (V.V.); ogoker-alpan@ldrtc.org (O.G.-A.)

**Keywords:** Fabry disease, enzyme replacement therapy, agalsidase, infusion-related reactions

## Abstract

Fabry disease (FD) is an X-linked lysosomal disorder caused by mutations in *GLA* gene resulting in lack of or faulty α-galactosidase A (α-GalA) enzyme. Enzyme replacement therapy (ERT) with recombinant human α-GalA enzyme (agalsidase) is the standard treatment option for FD. Infusion-related reactions (IRRs), with symptoms ranging from rigors, to fever, pain, vomiting, angioedema and diarrhea, are often seen due to immune response against the exogenous enzyme. To elucidate the mechanisms causing the IRRs in FD, eight patients who developed IRRs were investigated. All, except one, tested negative for agalsidase-specific IgE and had normal tryptase levels. Circulating dendritic cells were drastically reduced during IRRs, suggesting possible sequestration to the sites of inflammation. An increase in NK cells and a decrease in T cells were also observed. Cytokines IL-4, IL-8 and TNF-α showed a significant increase, indicating nonspecific degranulation of mast cells. All IRRs were managed successfully using a combination of standard premedications and mast cell stabilizers without any interruption of therapy. Taken together, the results indicate crosstalk between immune cells resulting in IgE-independent mast-cell-specific allergic inflammation. Mast cell stabilizers could be used to control IRRs and for safe reintroduction of agalsidase in patients previously treated with ERT.

## 1. Introduction

Fabry disease (FD) is a rare genetic lysosomal storage disorder caused by mutations in the *GLA* gene (OMIM#300644), and is inherited in an X-linked manner. It leads to a lack of or faulty α-galactosidase A (α-GalA) enzyme causing accumulation of the glycosphingolipid, globotriaosylceramide (GL-3) and its derivative globotriaosylsphingosine (lyso-GL-3) in lysosomes of several tissues and organs causing progressive damage that could lead to multi-organ failure involving kidneys, the heart and the central nervous system [1,2]. Enzyme replacement therapy with recombinant enzymes is the standard of care of treatment in Fabry disease (FD). Currently, two recombinant enzymes—agalsidase alfa (Replagal^®^, Takeda Pharmaceutical Company, Ltd., Cambridge, MA, USA) and agalsidase beta (Fabrazyme^®^, Sanofi Genzyme, Cambridge, MA, USA)—are available for patients with FD in the European Union, while only agalsidase beta is approved for use by US Food and Drug Administartion (FDA) in the USA [3,4,5]. Enzyme replacement therapy (ERT) with either of the recombinant enzymes has proven to be successful in mitigating the pathological effects and improving the quality of life in FD patients. However, infusion-related reactions (IRRs) are often seen in some FD patients as a result of immunogenicity of infused exogenous enzyme [6,7]. In clinical trials, 55% of patients who received algasidase beta at a dose of 1 mg/kg had experienced IRRs, some of which were severe [8]. According to Fabry Outcome Survey, most adverse events were mild IRRs, occurring in approximately 13% of patients on agalsidase alfa administered at a dose of 0.2 mg/kg [9,10]. It has also been noted that IRRs occurred much more frequently in male patients for both products. In most of the affected patients, IRRs occurred after the initiation of treatment. Subsequent generation of antibodies in patients with no residual α-GalA activity can cause significant morbidity, leading to interruptions and occasional discontinuation of therapy. Interestingly, IgE antibodies usually associated with type 1 hypersensitivity reactions are often not found in FD patients with IRRs [10,11], suggesting that IgE-dependent immune pathways are not the only culprit for the most IRRs in FD. The mechanisms and underlying immune perturbations resulting in hypersensitivity to infused enzyme are not yet fully understood. In an attempt to better elucidate the role of immune system and IgE-independent mechanisms in IRRs in FD patients, we analyzed peripheral blood drawn pre- and post-infusion from eight FD patients experiencing IRRs and compared it to FD patients who tolerate the ERT. 

## 2. Results

### 2.1. Infusion Related Reactions in Fabry Disease Patients during ERT

Eight male patients with FD developed hypersensitivity reactions during infusion of agalsidase beta, with symptoms ranging from rigors, to fever, pain, vomiting, diarrhea and angioedema appearing within a few minutes to hours after the start of infusion. Two subjects (ID#02 and 08) developed IRRs within 3 months of initiating ERT. Six subjects were under continued ERT for 2–5 years before they developed IRRs. The *GLA* pathogenic variants, IRR symptoms, complement analysis and NCI-CTCAE (National Cancer Institute Common Terminology Criteria for Adverse Events) grade are summarized in Table 1. Six patients presented with NCI-CTCAE grade 3 and two patients with grade 2 criteria for hypersensitivity and acute infusion reactions. All the subjects were male, and all but one had normal tryptase levels. Complement abnormalities were not seen in six patients, while two patients showed only reduced C4 level. Six subjects had neutralizing antibodies (Nab) with titers ranging from 1:20 to 1:500, while two did not have any Nab. All subjects were found to be positive for anti-agalsidase antibodies (ADA) of IgG type with titers ranging from 160–20,480. Skin testing (prick and intradermal) yielded positive results intradermally only in the subject with anti IgE ADA. No correlation was observed between the severity of FD symptoms, genotype, tryptase level, antibody titers and severity of IRRs. IRRs were managed using a combination of premedications that included corticosteroids, mast cell stabilizers, H1 and H2 blockers and IV fluids.

### 2.2. Dendritic Cells Sequestered from Circulation during IRR

Immunophenotyping analysis was performed to analyze T and B lymphocytes; their subsets; NK, NKT and dendritic cells (DCs) in FD subjects with IRR (reactors; *n* = 8); and those who tolerate the ERT (non-reactors; *n* = 9). There were no significant differences in overall immune cell fractions between reactors and non-reactors from the peripheral blood samples drawn just prior to infusion. To assess if specific immune changes occur within the FD patients during IRRs, the immune profile from samples drawn pre and post infusion were compared. Dendritic cells in the peripheral blood were detected as cells that are lineage-negative, CD34-negative and HLA-DR-positive. DCs were further sub-divided into myeloid and plasmacytoid cells based on surface expression of CD11c and BDCA2 (Figure 1A). In FD patients with IRRs, drastic decrease in DCs was observed post infusion (*p* = 0.0027). However, when similar comparison was performed in non-reactors, there was no change in DCs post infusion. In reactors, DCs were sequestered away from circulation in post-infusion samples when compared to non-reactors where no alterations in DCs were observed in pre- and post-infusion samples. Further, in reactors, the decrease was found to be due to myeloid DCs (*p* = 0.0028) (Figure 1B,C and Figure 2A–F).

#### 2.2.1. Alterations in T Lymphocytes and Natural Killer Cells

When immune profiles were compared between pre- and post-infusion samples, there were no changes in B lymphocytes and NKT cells in all FD samples. However, in reactors, a significant decrease in overall T lymphocytes (*p* = 0.011) and a corresponding increase in NK cells (*p* = 0.011) were found in post-infusion blood samples compared to pre-infusion. Similar alterations in T lymphocytes an NK cells were not observed in non-reactors (Figure 2H,J). All these immune alterations in DCs, NK cells and T lymphocytes were found to be immediate effects of IRRs, and were found to be reversible since their numbers normalized soon after the IRRs were stabilized.

#### 2.2.2. Non-Specific Mast Cell Activation Resulting in IRRs

Mast cells are known to be the effectors of type 1 hypersensitivity. Upon being triggered by various stimuli, mast cells release granules and powerful chemical mediators, such as histamine, cytokines, granulocyte macrophage colony-stimulating factor (GM-CSF), leukotrienes, heparin and many proteases. These chemical mediators cause the characteristic symptoms of allergy. In order to study if the hypersensitivity reactions were a result of mast cell activation, some of the known cytokines released as a result of mast cell degranulation were evaluated. Interleukin-4 (IL-4), IL-8 and TNF-α were quantified in plasma from pre- and post-infusion blood samples from subjects with IRRs and compared to FD subjects who tolerate ERT. Cytokines IL-4, IL-8 and TNF-α are significantly elevated (*p* = 0.012, 0.033 and 0.0291, respectively) in FD subjects with IRRs when compared to control group (Figure 3A–C), indicating the degranulation of mast cells upon exposure to recombinant protein. 

## 3. Discussion

Enzyme replacement therapy with rh α-Gal A has been available for the treatment of FD since 2001 in Europe and 2003 in the USA. Currently, there are two recombinant enzymes that are available: Agalsidase alfa (Replagal^®^, Takeda Pharmaceutical Company, Ltd., Cambridge, MA, USA) is administered at the dose of 0.2 mg/kg, and agalsidase beta (Fabrazyme^®^, Sanofi Genzyme, Cambridge, MA, USA) is administered at the dose of 1 mg/kg body weight [8,9]. Both the preparations are available for treatment in European Union and other countries, while only agalsidase beta is approved in the USA. Treatment with either of the recombinant enzymes has been shown to be beneficial, as shown by the significant reduction in plasma and urine GL-3 levels as well as plasma lyso-GL3 levels [5]. However, IRRs are often seen in FD patients as a result of immunogenicity of infused exogenous enzyme after the initiation of ERT. Most of the affected FD patients develop tolerability to infused enzyme, and the IRRs normally disappear after the first few infusions [10]. 

All FD patients in the current study who developed IRRs began to show symptoms within a few minutes of starting of the infusion, indicating immediate hypersensitivity reactions that result from mast cell degranulation resulting in allergic manifestations. Mast cells are long-lived tissue-resident innate immune cells that play a vital role in several inflammatory responses, including host defense to parasitic infection and in allergic reactions. Most often mast cells are activated in IgE-dependent mechanisms and produce and release various mediators; however, they can also be activated in IgE-independent manner [12]. It has been noted by several groups that IgE antibodies have not been detected, ruling out IgE specific hypersensitivity reactions [7]. Similar to observations from earlier studies on IRRs in FD patients, all eight patients were negative for IgE antibodies. Hence, we hypothesize that IgE-independent non-specific mast cell activation is present in IRRs in FD patients. Type II hypersensitivity reactions are known to occur due to abnormal activation of complement system [9]. In the current study, six subjects had normal complement levels, while two patients showed reduced C4 level, suggesting that primary cause for IRRs is not complement-mediated cytotoxic hypersensitivity reactions.

Studies have shown persistent immune cell perturbations in FD patients compared to normal controls, including elevated number of activated T cells, memory T cells and NK cells [13]. While immune profile showed no differences in major immune cell types, including T cells, dendritic cells and NK cells between reactors and non-reactors, when pre- and post-infusion samples were compared, circulating dendritic cells were drastically reduced in reactors while being unaffected in non-reactors, suggesting sequestration to the sites of inflammation. Bidirectional crosstalk between NK cells and DCs has been noted in certain inflammatory responses and allergic reactions. NK cells also play a role in the regulation of the adaptive immune response, and have been shown, in different contexts, to stimulate or inhibit T cell responses. When it comes to the hypersensitivity reactions to drug and other infused proteins, knowledge about the underlying mechanisms is limited. Drugs and recombinant proteins are thought to interact differently with dendritic cells from allergic and nonallergic patients, modifying their maturation level. Dendritic cells are also able to metabolize the infused biological substances and to present their metabolites to T cells resulting in a hypersensitivity response [14,15,16,17]. It has been shown earlier that mast cell degranulation can activate dendritic cells to facilitate inflammatory responses. Toll-like receptor (TLR) pathways may play a role in mediating interactions between DCs, T lymphocytes and mast cells, thus modulating allergic immune responses [18]. Even though IgE antibodies were not found in the FD patients with IRRs, we wanted to investigate if mast cells are being activated in IgE-independent manner leading to degranulation and resulting in secretion of inflammatory cytokines. Known mediators of mast cell degranulation—IL-4, IL-8 and TNF-α—were quantified in plasma from pre- and post-infusion blood samples, which showed significant increase of these cytokines specifically in FD patients with IRRs, indicating nonspecific degranulation of mast cells in response to exogenous recombinant protein. Taken together, the results of this study indicate IgE-independent activation of mast cell degranulation, resulting in activating DCs and cross-talk with other immune players, including NK cells as a possible mechanism for inflammatory responses causing IRRs in FD subjects (Figure 4). 

Mast cells not only participate in allergic immune response in various tissues and organs, including skin, gastrointestinal (GI) and pulmonary tracts, but also participate in other inflammatory diseases. The role of mast cell has recently been implicated in inflammatory disorders such as inflammatory bowel disease, metabolic bone disease, obesity and diabetes [19,20]. Cromolyn and ketotifen, two commonly used pharmacologic mast cell stabilizers, improved body weight and insulin tolerance in different strains of mast cell-deficient mice, suggesting the role of mast cells in non-IgE-mediated inflammatory response [20]. In this cohort, all IRRs were successfully managed using a combination of premedications and mast cell inhibitors (including mast cell stabilizers, steroids and H1/H2 blockers) without any interruption of therapy. ADA titers were noted to decrease upon repeated exposure once ERT is tolerated. Mast cell stabilizers could thus be used to control nonspecific mast cell activation associated with IRRs. Further studies on such interactions might help in predicting the clinical outcome and developing strategies to target such interactions to counter IRRs. 

## 4. Materials and Methods 

### 4.1. Subjects

Eight subjects with confirmed diagnosis of FD (all male) that showed drug-induced hypersensitivity reactions as well as nine FD subjects who tolerated ERT were identified from the subjects enrolled into IRB approved clinical study (NCT01745185, date of approval: 25 September 2012). Age and gender matched FD subjects who do not show infusion-associated reactions were enrolled as control subjects (non-reactors). Subjects were not diagnosed with concurrent autoimmune diseases. The handling of tissue samples and patient data was approved by the internal review board (Western IRB), including the procedure whereby all patients gave informed consent to participate in this study. Written informed consent was obtained using IRB-approved informed consent form. At enrollment, a medical history was obtained and a detailed physical examination was performed. Medical records were reviewed as a part of the clinical evaluation. 

### 4.2. Antidrug Antibodies and Neutralizing IgG Antibodies Titration

The titration for ADA and Nab was performed as described earlier [13]. Briefly, samples were first screened for ADA using an electro-chemiluminescent (ECL) bridging assay. Samples that screened positive were then confirmed by competition with agalsidase in the ECL bridging format. The titer of confirmed positive samples was determined using the same ECL bridging assay, and all confirmed positive samples were further characterized using an enzyme activity based Nab assay. Minimum required dilution for both the ADA and the Nab assay was 1:20.

### 4.3. Immunophenotyping

Direct immunofluorescence with specific antibodies was performed either on peripheral blood as previously described with some modifications using the following antibodies: anti-IgG1 FITC, anti-CD34-FITC, anti-IgG1-PE, anti-CD3-FITC/CD16+CD56-PE, anti-CD11C-PE, anti-CD21-PE, anti-CD20-PerCP and anti-HLA-DR-PerCP (BD Bioscience, San Jose, CA), anti-CD19-FITC, anti-CD3-APC (Invitrogen, Carlsbad, CA, anti-Lineage-FITC (anti CD3/CD14/CD16/CD19/CD20/CD56), anti-CD45-APC (Biolegend, San Diego, CA, USA) and anti-BDCA2-APC (Miltenyi Biotech, San Diego, CA, USA) [13,21]. Briefly, after washing the whole blood with PBS, 100ul of blood was stained with the relevant cocktail of antibodies at 4 °C for 30 min followed by red blood cell lysis using BD FACS lysis solution (BD Bioscience, San Jose, CA, USA). Samples were acquired on Accuri C6 flow cytometer (BD Bioscience, San Jose, CA, USA) and analyzed using FCS express software (De Novo software, Glendale, CA, USA). During acquisition, a lymphocyte gate was assigned, and 10,000 events were collected for the T cells and NK cells. For DCs, a million ungated events were acquired. 

### 4.4. Cytokine Analysis

Quantification of cytokines, IL-4, IL-8 and TNF-α (Thermofischer Scientific, Waltham, MA, USA) was performed from plasma samples collected from FD patients using ELISA as per manufacturer’s protocols.

### 4.5. Statistical Analysis

All statistical analyses was performed using GraphPad Prism software (GraphPad Software, Inc., La Jolla, CA, USA). Statistical evaluation of differences was performed using paired *t*-test to compare results from pre- and post-infusion samples from same subjects. *p*-values were indicated where found significant, * *p* < 0.05; ** *p* < 0.01.

## Figures and Tables

**Figure 1 ijms-21-07213-f001:**
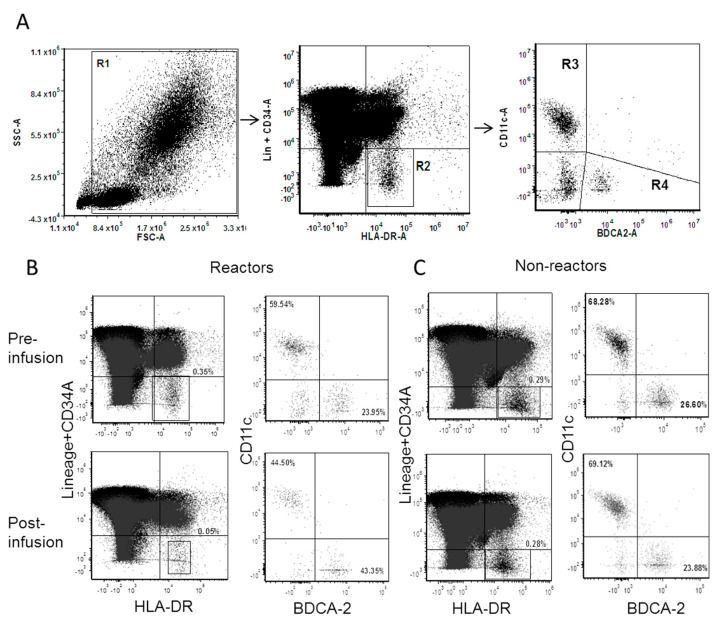
Flow cytometric analysis to detect dendritic cells (DCs) from peripheral blood: the figures are representative of typical patients. (**A**) R1 selects for white blood cells (WBCs). Dendritic cells are detected in R2 as the population of Lin− CD34−/HLA-DR+ and divided into Myeloid DCs− CD11c+ BDCA2− (R3), and plasmacytoid DCs− CD11c− BDCA2+ (R4). (**B**,**C**) Representative DCs, myeloid DCs and plasmacytoid DCs from Fabry disease (FD) patients with IRRs and no IRRs, respectively.

**Figure 2 ijms-21-07213-f002:**
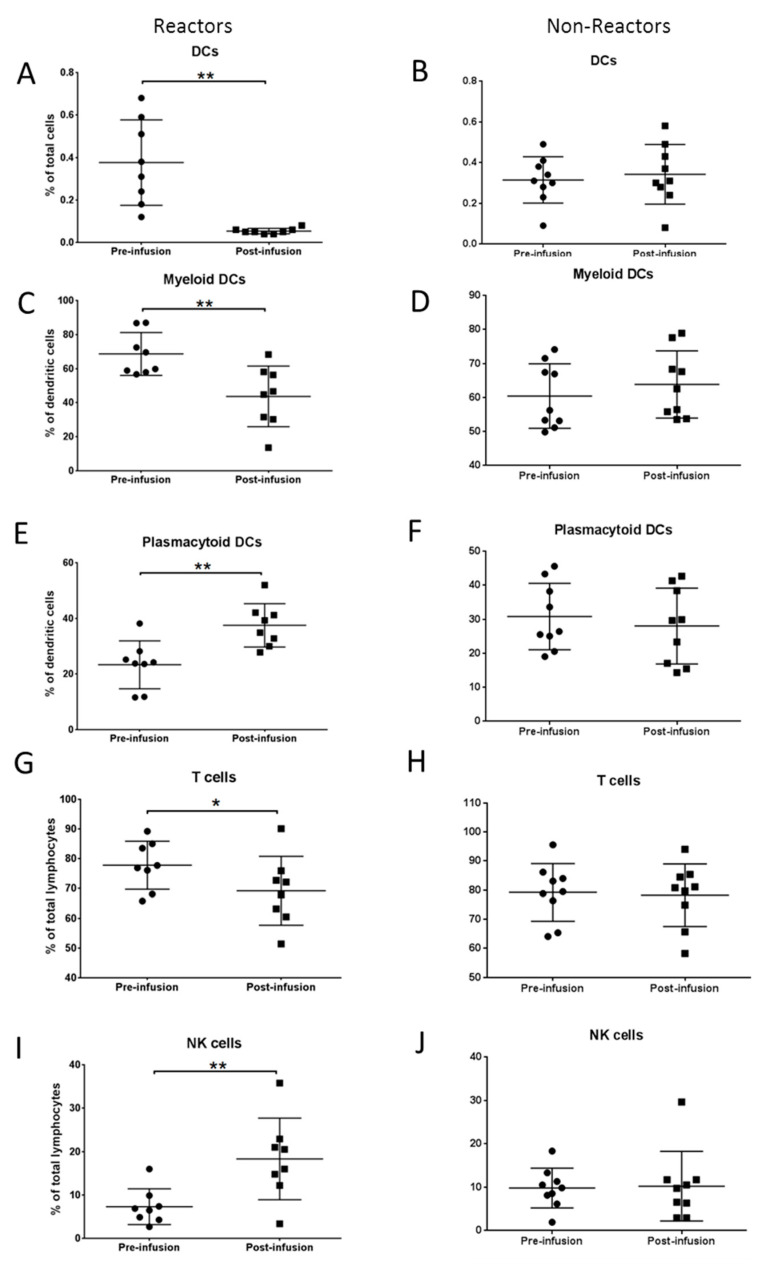
Immunophenotyping at pre- and post-infusion time points from FD patients with IRRs (reactors) and without IRRs (non-reactors). (**A**–**F**) Significant differences in total dendritic cells and subsets from circulation were observed in reactors compared to non-reactors. (**G**–**J**) Corresponding differences in circulating T cells and NK cells were also observed. *: *p* < 0.05; **: *p* < 0.01.

**Figure 3 ijms-21-07213-f003:**
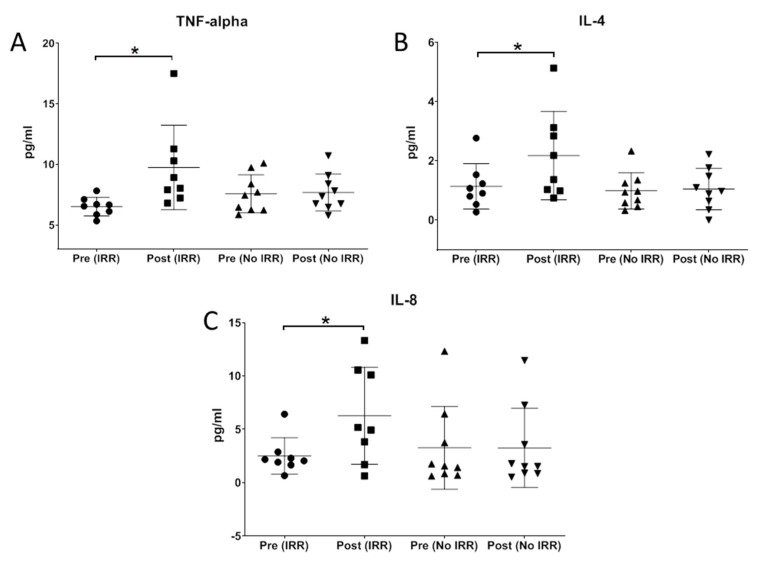
Cytokine analysis pre- and post-infusion time points from FD patients with IRRs and without IRRs. (**A**–**C**) Quantification of TNF-α, IL-4 and IL-8 in plasma samples at pre- and post-infusion time points reveals significant increase in subjects with IRR, compared to subjects with no IRRs. *: *p* < 0.05.

**Figure 4 ijms-21-07213-f004:**
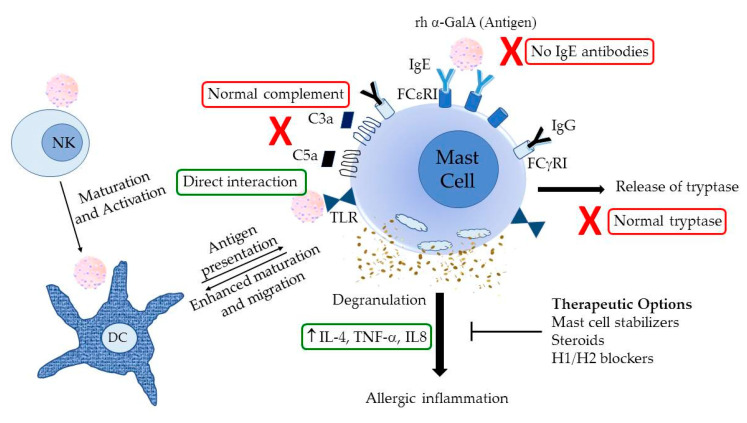
Possible mechanism of mast cell activation in IRRs in Fabry disease patients. NK, Natural killer cells; DCs, dendritic cells; TLR, toll-like receptor; Ig, immunoglobulin; rh α-GalA, recombinant human α-galactosidase A (agalsidase); IL, interleukin; TNF, tumor necrosis factor.

**Table 1 ijms-21-07213-t001:** Eight subjects with Fabry disease show infusion-related reactions (IRR) during enzyme replacement therapy.

ID	Sex	Age	Genotype	Tryptase	Complement	NCI-CTCAE Grade	ADA	Nab	IRR Symptoms	IRR Management
01	M	49	c.717delAA	Normal	Normal	3	20,480	Neg	Rigor, increased BP, pain	CS, AH, IVF, MS
02	M	28	P205T	Normal	Normal	2	320	1:20	Rigor, fever, pain hands/feet	CS, AH, IVF
03	M	26	R220X	Normal	Normal	3	160	1:20	Hypotension, fever, rigor	CS, AH, IVF, MS
04	M	9	R49P	Normal	Normal	3	2560	Neg	Sudden SOB, Rigor, vomiting	CS, AH, IVF, Epipen
05	M	25	c.256delT	Normal	Normal	3	2560	1:100	Pain, flushing, nausea, diarrhea	CS, AH, IVF, MS
06	M	35	C223Y	Elevated	Normal	3	2560	1:100	Rigor, pain, Fever	CS, AH, IVF
07	M	15	E103X	Normal	↓C4	3	20,480	1:500	Pain, rigor, SOB	CS, AH, IVF, MS
08	M	8	E103X	Normal	↓C4	2	10,240	1:100	SOB, vomiting	CS, AH, IVF, MS

SOB: Shortness of breath, BP: Blood pressure, NCI-CTCAE: National Cancer Institute Common Terminology Criteria for Adverse Events, ADA: Anti-agalsidase antibody titer, Nab: Neutralizing antibody titer, CS: Corticosteroids, AH: H1&H2 blockers, IVF: Intravenous fluids, MS: Mast cell stabilizers.

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
