# Peer review of "The Interaction of Innate and Adaptive Immunity and Stabilization of Mast Cell Activation in Management of Infusion Related Reactions in Patients with Fabry Disease"

_ijms, 2020, doi:10.3390/ijms21197213_

Round 1

Reviewer 1 Report

This work presents a reasonable attempt to elucidate the role of immune system and IgE independent mechanisms in Fabry patients developing hypersensitivity reactions to ERT. I find the research well designed and described, however more detailed information on the patients included in the study should be added and  discussed - especially age, FD severity, concurrent autoimmune disease, other medicines taken and the advance of the ERT (was it the very first enzyme infusion or any sequential?), which might be risk factors for infusion related reactions.

Author Response

Response to Reviewer 1 Comments

Point 1: This work presents a reasonable attempt to elucidate the role of immune system and IgE independent mechanisms in Fabry patients developing hypersensitivity reactions to ERT. I find the research well designed and described, however more detailed information on the patients included in the study should be added and  discussed - especially age, FD severity, concurrent autoimmune disease, other medicines taken and the advance of the ERT (was it the very first enzyme infusion or any sequential?), which might be risk factors for infusion related reactions.

Response 1: We thank the reviewer for the kind words and valuable suggestions. The additional information suggested has been added to Table 1 (age, and type of medications given to manage IRRs) as well as in text, section 2.1. We found that FD severity did not have any effect on IRRs and this information has been added to the text, section 2.1. The information regarding the advance of ERT before the subjects developed IRRs has been added to text, section 2.1. None of the subjects had any concurrent autoimmune disease, and this fact is added to the text in section 4.1.

Reviewer 2 Report

This is an interesting, well-written manuscript that provides interesting new insights into the IRR response to Fabry disease treatment that occurs in a significant proportion of patients. The conducted analyses are scientifically sound (with the exception of 1 conceptual problem, see comments) and reported in a transparant way. 

I have only minor comments: 

- Important conceptual comment regarding the conducted analyses: Throughout the manuscript pre and post treatment patient samples are compared for (1) the group showing IRR, and (2) the control group. However, no direct comparisons are made between the control and the IRR group. The presence of significance in the IRR group and the absence of significance in the control group is not the same as a significant difference between control and IRR groups (but conclusions are drawn as if it is the same). Direct comparisons between IRR and control should be made, for example by comparing the pre groups with one another (no expected difference), and then the post groups (expected difference). Or, to account for variation between patients, the authors could also consider taking first the ratio per patient of post/pre, and comparing this ratio between IRR group and control.  

- The authors write L179: 'In this cohort, all IRRs were successfully managed using a combination of premedications and mast cell inhibitors (including mast cell stabilizers, steroids, and 180 H1/H2 blockers) without any interruption of therapy.' But I could not find this information described in the results.

Author Response

Response to Reviewer 2 Comments

This is an interesting, well-written manuscript that provides interesting new insights into the IRR response to Fabry disease treatment that occurs in a significant proportion of patients. The conducted analyses are scientifically sound (with the exception of 1 conceptual problem, see comments) and reported in a transparant way.

I have only minor comments: 

Point 1: Important conceptual comment regarding the conducted analyses: Throughout the manuscript pre and post treatment patient samples are compared for (1) the group showing IRR, and (2) the control group. However, no direct comparisons are made between the control and the IRR group. The presence of significance in the IRR group and the absence of significance in the control group is not the same as a significant difference between control and IRR groups (but conclusions are drawn as if it is the same). Direct comparisons between IRR and control should be made, for example by comparing the pre groups with one another (no expected difference), and then the post groups (expected difference). Or, to account for variation between patients, the authors could also consider taking first the ratio per patient of post/pre, and comparing this ratio between IRR group and control. 

 Response 2: We thank the reviewer for the kind words and valuable suggestions. We have used paired t-test since that would reflect the significance taking into consideration the pre- and post-infusion changes occurring in each individual. Thus, it would not be possible to compare between different cohorts. However, we do agree that the phrasing in the results section would wrongly imply that we were comparing both cohorts, while we only wanted to show that the pre- and post- alterations were only seen in reactors and not in non-reactors. To eliminate this confusion, we have re-phrased the results section 2.2 and 2.3.

Point 2: The authors write L179: 'In this cohort, all IRRs were successfully managed using a combination of premedications and mast cell inhibitors (including mast cell stabilizers, steroids, and 180 H1/H2 blockers) without any interruption of therapy.' But I could not find this information described in the results.

Response 2: We apologize for the oversight. The suggested information has now been added to the results section: Table 1 (age, and type of medications given to manage IRRs) as well as in the text, section 2.1.